# Technology for Improving Street Dog Welfare and Capturing Data in Digital Format during Street Dog Sterilisation Programmes

**DOI:** 10.3390/ani12152000

**Published:** 2022-08-08

**Authors:** Amit Chaudhari, George Brill, Indira Chakravarti, Tim Drees, Shrikant Verma, Nidhi Avinash, Abhinandan Kumar Jha, Sitaram Langain, Narendra Bhatt, Sanjit Kumar, Satyanarayan Choudhary, Parvinder Singh, Subhash Chandra, Anju Murali, Katherine Polak

**Affiliations:** 1Humane Society International, 1255 23rd Street NW, Suite 450, Washington, DC 20037, USA; 2Volunteer, Humane Society International, 13721 Nogales Dr, Del Mar, CA 92014, USA

**Keywords:** street dog, spay/neuter, rabies, dog welfare, phone apps, animal birth control

## Abstract

**Simple Summary:**

Humane Society International (HSI) facilitates dog sterilisation programmes internationally, which includes population surveys of street dogs to gain basic demographic information and to set a baseline for future monitoring operations. HSI has developed a web and mobile application suite called ‘HSIApps’ with custom tailored workflows to improve the efficiency (lower programmatic cost) and improve the welfare of dogs in care throughout the sterilisation process. The Android-based mobile app is simple and easy to use for teams in the field. The web app has data dashboards, record views, and reports for monitoring and evaluation purposes. The use of such digital applications can improve dog population management programme implementation, ensure positive outcomes for dogs postoperatively, and facilitate programmatic monitoring and evaluation. We describe here the use of this application and insights gained through its use in HSI’s street dog monitoring, evaluation, and impact assessment (MEIA) programmes in India.

**Abstract:**

Street dogs survive on food handouts provided by individuals, or the wider community yet typically receive limited to no veterinary care. They can also carry a variety of zoonotic diseases such as rabies, posing a significant risk to human and dog population health. Dog sterilisation is one of the most humane and effective methods available to control street dog populations. Dog sterilisation programmes, particularly those operating at a large-scale, often face a variety of challenges including limited resources, staffing, and less-than-ideal facilities. Recordkeeping is often a challenge as well, which can complicate the return of a sterilised dog to their location of capture. Street dogs are territorial, and the return of a dog to an incorrect location is fraught with various welfare issues, as well as an increased risk of postoperative complications, including death. Humane Society International developed a mobile phone-based application called ‘HSIApps’ drawing on years of field experience and data collection in street dog location recording, as well as clinical and postoperative treatment. HSIApps facilitates the return of dogs back to their exact captured location, which ensures dog welfare, and generates reports of a variety of useful data variables to maximise the efficacy and reliability of sterilisation programmes.

## 1. Introduction

### 1.1. Free Roaming Dogs

Dogs living on the street without immediate human supervision are known as free-roaming [1]. This definition includes owned dogs that are allowed to roam for prolonged periods, owned dogs that live on the street, and true street dogs. A street dog can be defined as one that is born on the street, survives on the street, and successfully produces progeny there [2]. India has a substantial free-roaming dog population, the majority of which are street dogs. There are an estimated 75.9 million dogs (55 dogs per 1000 humans) in India, both owned and unowned, 62.1 million (45 dogs per 1000 humans) street dogs, and 13.9 million (10 dogs per 1000 humans) owned dogs [3]. 

Street dogs live in harsh conditions with little to no veterinary care and are always at a high risk of road accidents. Street dogs are also under threat from various parasitic infections and zoonotic diseases, many of which represent a human public health concern, such as rabies. Together, these factors contribute to street dogs having a short average life expectancy in comparison to owned nonroaming dogs. Studies conducted on street dogs in India found that only 18% of pups survive a year [4], and that the average lifespan for a spayed female street dog is only 3.8 years [5]. Indian street dogs have one specific breeding cycle (monoestrous) beginning during the post-monsoon season from August to December, with peak pup ratios occurring in the following months (January to March). Street dogs are successful breeders and maintain the population at the carrying capacity unless decisive, significant intervention interrupts them. Indeed, the proportion of female street dogs becoming pregnant is estimated at approximately 47.5% in any given year [5], and models show that populations may persist at carrying capacity even despite low proportions of sterilisation and/or euthanasia intervention [6].

Street dog survival is largely dependent on direct or indirect food provisioning from people [7,8,9], who may provision regularly according to religious beliefs or out of kindness; feeding street dogs is culturally a routine practice in India for many households. According to HSI household surveys across India, between 58% and 95% of households feed street dogs, with frequency ranging from daily to once a month. 

### 1.2. Importance of Successful Sterilisation Programmes

Sterilisation programmes are an effective tool for controlling street dog populations. High densities of free roaming or street dogs increases the risk of human dog bites, zoonotic disease transmission, and poorer animal welfare for both domestic and wild animals [10].

Although there are more than 300 possible zoonoses that dogs can transmit to humans [11], the rabies virus is perhaps of highest concern. Rabies is endemic to India, with an estimated 18,000–20,000 cases in humans per year. A large portion of these cases are attributable to the more than 17.4 million annual dog bites in India; dogs represent the primary reservoir of the virus and are the source of more than 99% of all human rabies cases [12]. Other notable canine-borne diseases that affect humans include visceral leishmaniasis, echinococcosis, and toxocariasis [11].

Coexistence issues between local wildlife and domestic dog populations also arise when free-roaming dog populations are not controlled. Domestic dogs can transmit pathogens to wildlife populations that may be more susceptible to infection. For example, the rabies virus has been transmitted to endangered wild canids in Africa, such as the Ethiopian wolf [13] and the African wild dog [14]. A recent report showed that populations of the endangered Asiatic wild dog, or dhole, of India have also been infected with rabies [15]. Beyond disease transmission, domestic dogs can prey on wildlife or compete for local resources [16]. In the case of other canine species, free-roaming domestic dogs can lead to hybridisation [17].

Welfare of the domestic dog populations themselves is also a central driver for the implementation of management programmes. Conflict with humans due to excessive population density in urban areas puts street dogs under threat of removal, relocation, injury, and death from local people and authorities. The welfare of street dogs is mostly ignored in developing countries often due to a lack of animal protection laws, awareness of existing laws if laws do exist, and poor legal implementation. In India, the Animal Birth Control Rules of 2001 (ABC Rules 2001) is a specific law that provides legal protection to street dogs against relocation, removal, killing, and poisoning [18]. ABC Rules 2001 suggest that the only humane way to manage street dog populations is by adopting a “catch, sterilise, vaccinate, and return” method for population control. The Prevention of Cruelty to Animals act of 1960 also provides legal protection to animals in India from any type of cruelty [19].

Sterilisation is currently the most effective humane method available for controlling street dog populations when conducted in a systematic way. [20]. The same can be said for canine rabies vaccination programmes. Other more inhumane methods (killing, poisoning, and relocation) of controlling street dogs have proven less effective at managing dog populations and may even introduce rabies to new areas [21,22]. Street dog sterilisation programmes help improve the health and welfare of street dogs and reduce juvenile mortality by reducing breeding [23], and they may also reduce human dog-bite injuries by acting to remove cause for maternal protective behaviours and by reducing roaming dog density [24].

### 1.3. Barriers to Successful Dog Population Management (DPM) Programmes in India

Under the ABC Rules 2001, local government authorities interested in controlling local dog populations often contract with animal welfare organisations to implement sterilisation and vaccination programmes, with funding and under supervision of the local authority. Several municipal corporations and local urban authorities in India have implemented street dog sterilisation programmes in the last two decades, employing the ‘catch, neuter, vaccinate, and return’ (CNVR) method. However, the majority of programmes have been conducted non-systematically, without baseline assessment of the local dog population, and without estimation or understanding of the intervention scale required to interrupt the replacement rate of street dog populations, or the fertility rate required by female dogs to maintain the population. Such programmes often result in little to no noticeable changes in the dog population, from a size and/or welfare perspective. As a result, funding is often withdrawn given the lack of perceived or real impact, and such programmes may be criticised publicly in the media. While a 70% sterilisation rate is often proposed as a necessary threshold for effective dog population reduction (e.g., news article—Times of India), this is neither supported by theoretical models nor actual data, and support for this figure cannot be found in the literature. Instead, it may be that the sterilisation of more than 83% of fertile female dogs is critical to successfully interrupt the breeding cycle of the population and noticeably reduce the population [25]. Furthermore, the sterilisation of both male and female dogs with limited resources and infrastructure may increase the overall length of the programme. Prioritising females or only focusing on female dog sterilisation is the most cost-effective method and reduces the necessary duration of the programme [26]. This is supported by the female-centric Animal Birth Control programme guidelines presented by the Animal Welfare Board of India [27]. Sterilisation programmes are only effective if conducted systematically, covering small areas one after another and ensuring that the sterilisation rates for each area and collectively across a geographic area are very high (above 83%) and consistently maintained [27]. 

From an operational perspective, a street dog population management programme in India typically begins with the physical capture of dogs. This generally occurs during early morning hours to avoid human and traffic movement. A dog-catching team records basic details and the location or address of the caught dogs on paper, but it can be difficult to describe a specific location without technological assistance such as GPS. Once 10 to 15 street dogs (depending on the size and type of catching vehicle used) are caught, they are transported to the programme’s clinic location. Once at the clinic, dogs are generally unloaded, receive a medical examination, and undergo surgery if healthy enough. While dogs are under anaesthesia, one of the ear tips is given a ‘V’ shape cut as a part of permanent identification. Dogs are housed in kennels for three nights post-operatively under observation to ensure their recovery. Additional data recorded during this process often also gets recorded and remains on paper, although some programmes may enter data into a Microsoft Excel sheet. A final check of the surgical wound is performed by a qualified veterinarian before the dog is returned to their location as detailed in the catcher’s log. A commonly encountered issue, however, is that the paper record is often not sufficient to ensure accurate return of dogs to their original locations where they were caught. The use of paper records also precludes any possible cross-verification for the programme management team to know if the dogs are indeed returned to the appropriate location. 

Such programmes can also collect a variety of data throughout the sterilisation process including physical characteristics of the dog (sex, colour, weight, age group, female dog in heat, etc.), disease conditions, skin/ear/eye infection, surgery-related data (surgery length, pregnancy, conditions related to the uterus or testis, and postoperative complications), which can facilitate a better understanding of the local dog population, as well as strengthen the program. 

Street dogs are territorial and live in specific areas, actively attacking and repelling extraterritorial dogs [28], thus maintaining territories and preserving locally available food resources. Poorly implemented street dog sterilisation programmes often fail to ensure that sterilised dogs are returned to the same territory they were retrieved from, typically due to poor record keeping in the form of paper-based data systems. However, in recent decades, smart phones have become commonplace, and phone-based applications are playing an increasingly important role in many sectors, including animal science and data collection. The integration of a free, easy-to-use smartphone-based application could greatly enhance the data collection and management of street dog sterilisation programmes.

A smart phone application-based data collection and analysis system provides digital forms of datasets which are essential in the modern world, to both evaluate efficiency and assess programme strengths and weaknesses. Humane Society International has been promoting humane methods of dog population control worldwide and working in India with local and state governments since 2012. In the last decade, HSI-India has implemented evidence-based and data-driven street dog sterilisation and anti-rabies vaccination programmes in several cities and states across India. HSI invested in technology and developed HSIApps, an innovative smart phone-based application and a web-based backend dashboard based on several years of ground experience of programme implementation. HSIApps passed several development stages and reached a stable stage where it could serve as a technology platform for implementing dog/cat sterilisation programmes anywhere globally in an efficient manner. It has been developed such that it can handle more than the typical amount of data generated by large-scale programmes without limitations. A sophisticated application with an online dashboard provides technology power to the programme managers and enables accurate data gathering, real-time monitoring, and accurate, GPS location-based report generation. The secondary data collected by HSIApps can be used to gain insight and knowledge about the street dog population. 

## 2. Materials and Methods

### 2.1. Programme Sites

HSI operates dog population management projects of various sizes throughout India aimed at sterilising and vaccinating more than 83% of street dogs. These projects are conducted in collaboration and with the support of the local municipal councils. Project sites included in this paper include those in the cities of Vadodara, Lucknow, Dehradun, Dindigul, Nainital, Mussoorie, and Kodaikanal. Some of the programmes sterilise an average of 80 street dogs per day and care for a daily inventory of 200 dogs in their kennels (due to the mandatory postoperative observation period of 72 h), with staffing of up to 35 people.

All programmes operate according to the provisions of Animal Welfare Board of India (AWBI) and Animal Birth Control Rule of 2001, following their “catch, neuter, vaccinate, and release” protocols. 

The data collection periods and total number of dogs recorded in HSIApps for each city are shown in Table 1.

### 2.2. Technology and Data Recording

The technology developed by Humane Society International includes the HSIApps smartphone application and an associated back-end web-based dashboard. HSIApps is designed to be used in the field to collect data while catching street dogs for sterilisation, while the online dashboard acts as a cloud-based server to enable centralised project management and reporting. The current version of HSIApps is compatible with Android phones (an iOS-supported version will be made available in the future), and the online dashboard can be accessed using any internet browser. HSIApps can work without internet connection (offline), whereby data are stored within the phone and uploaded to the server upon reconnection to internet. HSIApps accesses phone location, allowing it to map data via GPS. A stable internet connection is required to record and save sterilisation clinic data entries on the online dashboard, and bulk data entries are allowed using the multi-record function.

HSI dog population management programmes employ staff serving in a variety of roles pertaining to catching dogs, performing surgery, caring for dogs post-operatively, and returning them. The HSI mobile phone application is integrated into the daily work of multiple staff members. 

A field officer or animal handler makes a new data entry for each dog captured using the ‘catch dog’ screen in HSIApps on a smart phone. It records dog details including sex, age group, and method of catching, as well as an image of the dog, and automatically fetch phone’s date and GPS location (Figure 1 and Figure 2) before the data are automatically uploaded to the central server for management access. HSIApps is designed to be user-friendly, straightforward, and intuitive, and it does not require technical skills beyond familiarity with smartphone operation. HSIApps replaces conventional paper-based methods of recording street dog caught locations, and images serve as additional verification during the sterilisation process and relocation identification.

The second step in a street dog sterilisation programme is the transfer of caught dogs to the clinic location for sterilisation, vaccination, and postoperative care. During this process, various details are gathered on a clinical case sheet (for example, physical examination data, surgery details, and postoperative care information). These details are transferred to the online web version of the application via manual entry into automatically synchronised entries related to each dog’s original capture data. Once the application has saved catch data and clinic data, the individual is visible under the ‘return’ feature of HSIApps, indicating return after the designated post-surgery observation period, assuming no complications occur. The ‘return’ button activates when the dog is marked as ready for return by a qualified veterinarian. Once dogs are marked for return, the application generates a navigation map to ensure each dog is accurately returned to their original capture location which is used by the dog-catching field team for accurate return. The return process is also closely monitored with a safety feature called ‘geofencing’. This geographical demarcation ensures accuracy of return to within 20 m of capture location. Upon reaching the location, a field officer presses the ‘return’ button on the phone screen; should the officer be outside the 20 m radius, the ‘return’ button will remain unavailable.

### 2.3. Online Dashboard

The web-based online dashboard is designed to support several aspects of street dog sterilisation programme management. It facilitates clinic data entry, return operations, programme monitoring through various automatic analyses, and programme data report generation. A project manager or admin user can log in to access the online dashboard, providing an overview of the entire project. The dashboard itself displays a combination of analytics representing project standing with respect to its specific targets and timeline. Through the online web application, one can view statistics such as the number of dogs sterilised, postoperative complications, average surgical time, and microchip number. Users provided with administrative access can edit the information on the online web dashboard and download programme data reports. The reporting feature is flexible with an option to download periodic, location-wise, brief, or descriptive data as required, in formats such as .xlsx (Microsoft Excel).

## 3. Results and Analyses

The data presented in this paper were collected through HSIApps. Where the cities represented possess additional data (such as data collected prior to HSIApps integration into the program), such data were excluded so as to demonstrate the capacity of the app most accurately. 

### 3.1. Catch-Site Data

Initial capture-site entry into HSIApps involved data on catch methods, age, sex, and physical characteristics. The methods used to catch dogs across the cities analysed there were as follows: hand, net, owner, scruff, and trap (Figure 3). Catching methods varied by site, with hand and net catching representing the most popular methods in most regions. In Kodaikanal, owners bringing in dogs for sterilisation represented the most common method. For each dog caught, age, sex, bodyweight, and coat colour were recorded. The ratio of the caught males and females varied from 0.08 to 1.19 (Table 2), while Figure 4 shows the proportion of puppies, male dogs, and female dogs caught and sterilised within each city, which varied for each location according to dog demographics and local government’s requirements. Finally, Figure 5, Figure 6 and Figure 7 show the distribution of dog bodyweight by age, sex and city, and colour distributions. Generally, the populations of dogs sterilised in each programme site were similar in composition with regard to age, sex distribution, bodyweight, and colour. More than 50% of the street dog population was found to have either brown or brown/white colours. The average body weight of dogs from hilly areas (Nainital, Mussoorie, and Kodaikanal) was higher compared to nonhilly areas (Vadodara and Lucknow). 

### 3.2. In-Clinic Data

Once dogs were brought into the sterilisation clinics, a variety of data variables were collected before, during, and after sterilisation. Initial health conditions were assessed, and incidences of the following were recorded: cryptorchidism, mono-orchidism, single uterus, ovarian cyst, pyometra, transmissible venereal tumours (TVTs), and mange. The distributions of conditions between locations are shown in Table 3. The proportion of pregnant and lactating females were also recorded, which, when combined with the proportions of pups at each location, provided insight into the breeding seasonality of the population in question. According to our samples, it appears that the street dog populations followed a seasonal breeding dynamic. This was discussed in detail in Brill et al., 2022.

Surgery duration and outcome details were also recorded in the app, allowing for in-depth analyses of surgery impacts and procedures. As an example of such possibilities, findings from the large-scale surgery datasets of Dehradun, Vadodara, and Lucknow are presented briefly here. 

Note that pregnant sterilisation surgeries were found to be longer in duration than nonpregnant sterilisation surgeries (Mann–Whitney U test: W = 52,889,700, *p*-value < 0.001). As such, unless otherwise specified, all surgery data involving pregnant females were removed to ensure technical equivalence of operation. Similarly, statistical analyses were separated by sex due to the difference in their nature of surgical operation. 

Combined, the distribution of surgery durations (pregnant dogs excluded) in Dehradun, Vadodara, and Lucknow (*n* = 60,314) was as follows: 50% of all surgeries took less than 10 min, 75% of all surgeries took less than 14 min, and 95% of all surgeries took less than 23 min. Location- and sex-specific surgery durations, as well as their post-operative incidence rates, are presented in Table 4.

On average, dogs that experienced postoperative complications had undergone longer surgeries that those that did not have complications. For female dogs that died post operation, these had undergone longer surgeries than those that did not; this was not evident in male dogs (see Table 5). Concerning sex groups, we found that increases and decreases in postoperative death and complications were significantly associated with sex (*p* < 0.001 and 0.008, respectively; see Table 6). Females experienced a significantly higher proportion of deaths, whereas male dogs experienced a significantly higher proportion of postoperative complications. Within sexes, the only significant association of age with postoperative fate was that of post-operation complications in female puppies, which were significantly more likely to experience post-operation complications than female young or adult dogs (post hoc test: χ^2^ = 3.824; *p* < 0.001; Table 7).

Lastly, according to our data, the sterilisation of pregnant females was no more or less likely to result in post-operation death or complications than the sterilisation of nonpregnant females (χ^2^ = 1.7728, df = 1, *p* = 0.183 and χ^2^ = 1.6142, df = 1, *p* = 0.2039 respectively; see Table 8). See the discussion for the relevance of this analysis to HSI’s sterilisation of pregnant females.

Recording the name of the surgeon performing each surgery also enabled analyses to be carried out concerning the effects of experience on surgery durations and outcomes. For example, as expected, inexperienced (trainee) veterinarians were found to take significantly longer on average to perform surgeries than qualified veterinarians (Mann–Whitney U Test: W = 8,131,730, *p*-value < 0.001; see Table 9). No postoperative deaths occurred under trainee-led operations, whereas 0.17% of deaths happened under qualified surgeons. However, this is likely due to more complex cases with a higher risk of complications being given to more qualified surgeons (see also discussion). We also acknowledge our small sample size for trainee operations. When testing for differences in postoperative complication rates between vets and trainees, we conclude that trainee vets were responsible for a significantly higher rate of postoperative complications than qualified vets (chi-squared test: χ^2^ = 35.244, df = 1, *p*-value < 0.001; see Table 9), as we might expect.

We also found that total the number of operations performed by the surgeon was significantly correlated with the number of postoperative deaths and complications. Figure 8 indicates that, on average, surgeons who completed more operations exhibited a lower rate of postoperative complications (rho = 0.586, S = 188.24, *p* = 0.0277) and deaths (rho = 0.838, S = 73.811, *p* < 0.001). However, note that operations performed prior to, or outside of, the HSI clinic records were not recorded. For these final specific analyses, trainee vets were excluded, since the small number of operations performed by trainees may have led to erratic postoperative death or complication percentages (with the result of each individual occurrence resulting in a comparatively large percentage difference). The larger (750+) number of total operations performed by each qualified vet ensured that the postoperative percentages were reliable for correlation analyses. 

## 4. Discussion

The methods and data we presented in this study showed the successful use of the HSIApps app for implementation of dog sterilisation programmes and associated data collection. The advantages of such a platform and the specific insights gained from the exemplary evaluations of such a dataset (as obtained during HSI’s India programmes) are examined below.

### 4.1. HSIApps Role in Returning Street Dog to Caught Location, Digital Data Gathering, and Visualisation of Sterilisation Coverage on Google Maps

In India, street dogs are protected by law against relocation and harmful interventions. Street dogs live free-roaming among human habitation, associated with defined territorial ranges [28]. By facilitating the accurate return of street dogs, HSIApps solves one of the core problems associated with street dog management programmes—that of territorial dislocation. The digital data gathered using HSIApps not only allows for various data analyses, but also improves the credibility of the programme by ensuring the return of dogs to their correct locations; this is important particularly when such programmes involve multiple stakeholders. Additionally, digital data allow for easy sharing, easy verification, and cross-organisational collaboration to reach high sterilisation rates—greater than 83% of females [25]—in each area for successful reduction in street dog population density. The online dashboard of the application with in-built Google Maps function creates a visualisation of sterilised dog’s locations on the map, providing programme managers a real-time understanding of the street dog sterilisation coverage in order to adjust resource allocation as necessary. 

The postoperative complications and deaths were significantly low for spay/neuter programmes reported here. At maximum, only 0.23% of adult females and 0.1% of adult male dogs died post-surgery (Table 7). Young and adult male dogs were more likely to have postoperative complications than similar categories of female dogs, although the survey of female dogs (spayed) is much more complicated and requires an open abdomen cavity for removal of the uterus and ovaries (Table 7). 

### 4.2. Female Dogs Are More Likely to Die Post Operation Than Male Dogs, but Pregnancy in Females Does Not Make Them More Likely Than Nonpregnant Females to Die Postoperatively

We examined the occurrence of postoperative death and complications in sterilised dogs. Complications in female dogs after an ovariohysterectomy (OVH) include haemorrhage, wound healing complications, ovarian remnant syndrome, stump pyometra, uretal injury, bowel obstruction, and acquired urinary incontinence [29]. Complications of orchiectomy in male dogs include haemorrhage and inadvertent prostatectomy. We found that a dog’s sex was significantly associated with postoperative complications and death, with females being more likely to experience postoperative death and males being more likely to experience postoperative complications. We suggest that this is because the complications inherent to female sterilisation surgeries are more likely to lead to death than those of male sterilisation surgeries, as supported by previous studies [30].

Our data show that sterilisation of pregnant females is no more likely to result in postoperative death or complications than sterilisation of nonpregnant females. This is despite the fact that the greater mean duration of pregnancy sterilisations has the potential to lead to a higher proportion of post-operation deaths and complications on average given the link between surgery duration and postoperative complications and death. The small sample size of pregnant complications should, however, be acknowledged; further research is required to confirm this comparative indication.

It should be noted that HSI’s protocol is to avoid capturing visibly pregnant female dogs for sterilisation. However, there are instances where a female dog is only identified as pregnant once on the operating table at the point of sterilisation operation. In such cases, the female dog is spayed as for nonpregnant females, by removal of the uterus with both the ovaries (ovariohysterectomy), along with all foetuses.

### 4.3. Younger Females Are More Likely to Have Postoperative Complications

Age of females was a significant predictor of postoperative death and complications, with female puppies (individuals less than six months of age) more likely to experience postoperative complications. There are many considerations, both long- and short-term, in determining the optimal age to spay a female dog. Long-term health considerations include longevity, orthopaedic disorders, and mammary neoplasia [31]. In the short term, the main risks stem from infections, abscesses, ruptured surgical wounds, and chewed out sutures. However, the relationship between age of female dog and presence of postoperative complications is contested, and conflicting evidence exists [32]. Traditionally, many veterinarians choose to spay female dogs after the second oestrus cycle or before 2.5 years of age [33]. Today, research suggests that the breed of dog should also be considered; dogs of different breeds mature at different rates and may, therefore, have different optimal spaying ages [34]. It is possible that some females spayed as part of this sterilisation programme were not spayed at the optimal time. Further research into similar large-scale datasets could provide further insight into the optimal age of female surgeries.

### 4.4. Surgeon Experience Directly Correlates with Postoperative Outcome

No trainee vets were responsible for any postoperative deaths in this dataset, while the qualified surgeons saw 0.17% of all their surgeries end in death. We propose, however, that this was due to the complex operation cases with more probable mortality being given to qualified vets rather than trainees, as well as the considerably smaller sample of trainee operations in the dataset. Indeed, a similar percentage of deaths (0.17%) among trainees would equate to less than a single postoperative death (0.46 to be precise) within the 272 operations performed by trainees in this dataset. Whilst a greater number of postoperative complications means we can calculate an incidence rate for these, the potential unreliability of small percentages for our lesser trainee sample size must be acknowledged.

As would be expected, trainee vets on average took longer to complete surgeries. Such dataset analyses present programme-specific insights that may be used to inform the most efficient allocation of surgeries among the vets available, i.e., the designation of more complex cases to more trained surgeons whilst allowing trainees to complete more straightforward surgeries. The data suggest that this is likely already being applied in this sample, but a more specific, tiered system could be put into place. In such a system, an initial evaluation based on certain dog criteria could determine the surgery complexity and operations be assigned accordingly.

## 5. Conclusions

The HSIApps mobile phone application and associated online dashboard developed by HSI show great potential in improving the execution and monitoring of dog population management programmes. Improved monitoring and evaluation are powerful tools in programme targeting and efficiency maximisation. The app also has major implications for relocation-related welfare concerns and as a platform for large-scale data collection and associated analysis. In particular, the HSIApps GPS integration provides a new layer of data recording and functionality previously unavailable. We presented here a successful pilot study of the HSIApps app, described its fundamental monitoring and evaluation functionality, and demonstrated its potential use in dog population management worldwide.

## Figures and Tables

**Figure 1 animals-12-02000-f001:**
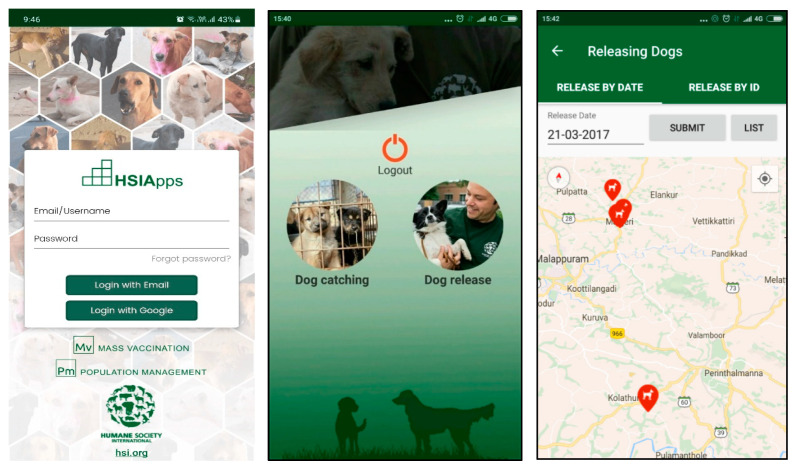
Screenshots of the HSIApps application, showing main login screen, catch–release screen for spay/neuter, and street dog return screen from left to right.

**Figure 2 animals-12-02000-f002:**
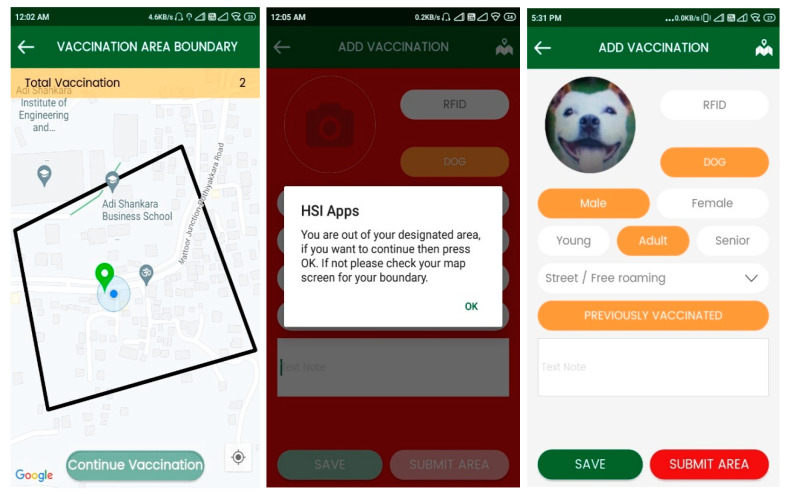
Screenshots of the HSIApps mass vaccination application, showing geofencing of an area screen, notification screen, and vaccination screen from left to right.

**Figure 3 animals-12-02000-f003:**
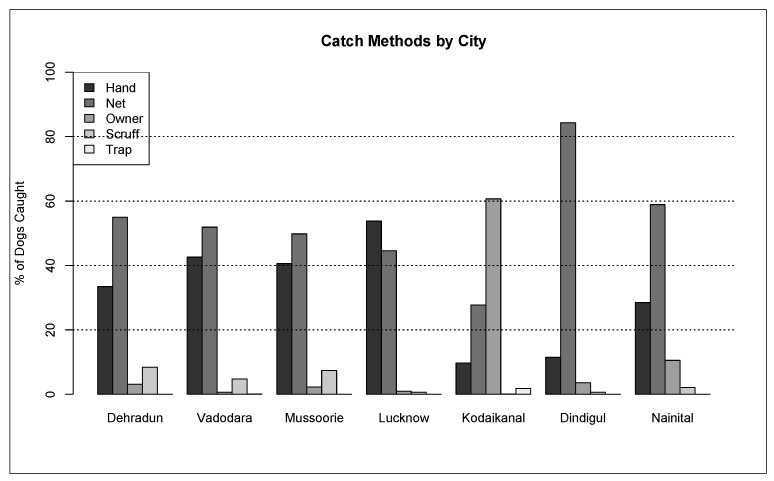
Street dog catch methods across programme sites.

**Figure 4 animals-12-02000-f004:**
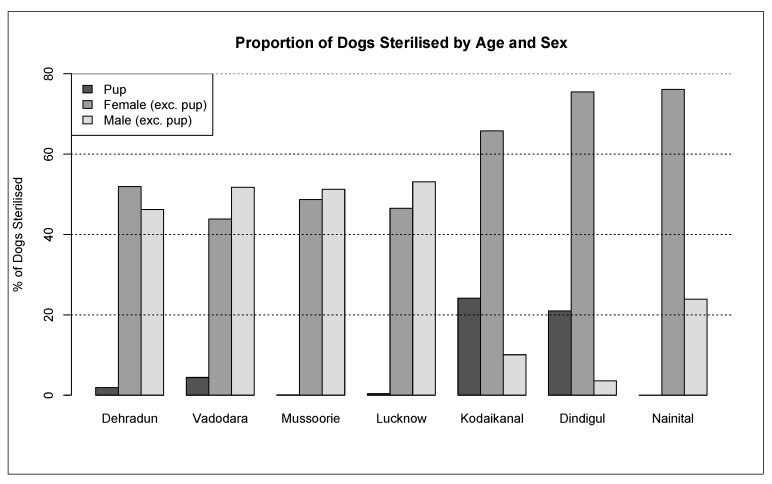
Proportion of dogs sterilised by age, sex, and city.

**Figure 5 animals-12-02000-f005:**
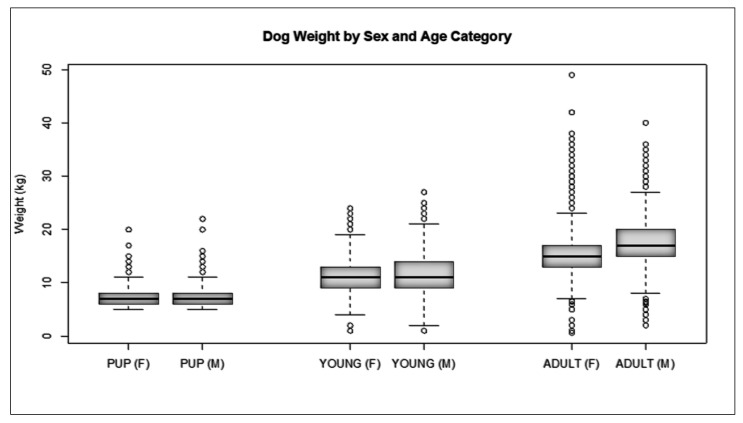
Dog weight by sex and age category across all sites.

**Figure 6 animals-12-02000-f006:**
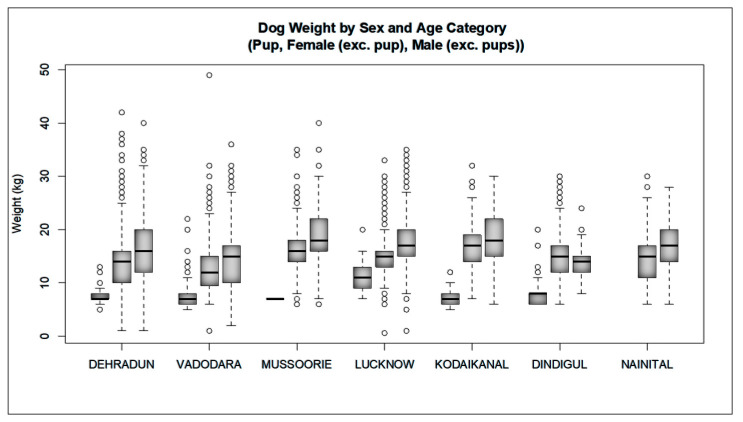
Dog weight by age and sex category at each programme site.

**Figure 7 animals-12-02000-f007:**
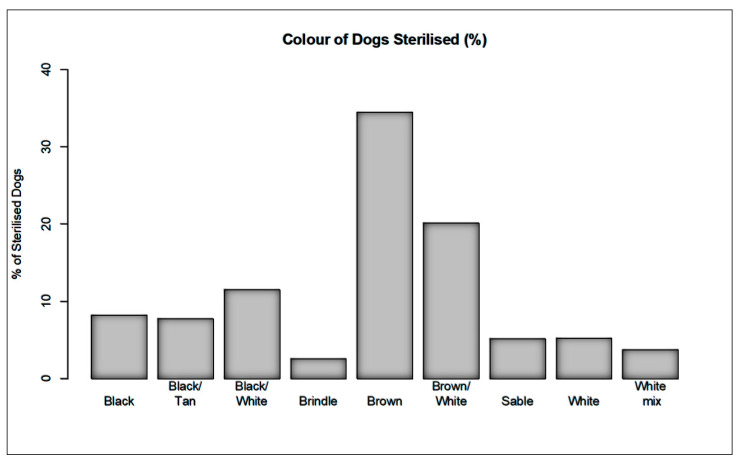
Street dog body coat colour recorded at each programme site.

**Figure 8 animals-12-02000-f008:**
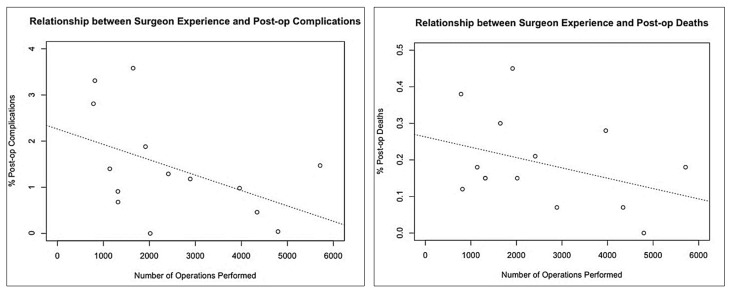
Relationship between total surgeries per surgeon and postoperative complications and deaths.

**Table 1 animals-12-02000-t001:** Data collection periods for programme sites.

Project Location	Data Collection Period	Period When Data Were Not Collected	Months of Data Collection	Data Collection Ongoing?	Total Dogs Sterilised
Dehradun	Jan 2018–Jul 2021	Apr 2020, May 2021	41	Yes	20,291
Vadodara	Sep 2017–Jul 2021	Apr 2020	46	Yes	19,496
Mussoorie	Sep 2018–Jul 2021	Jan–Jul 2019; Nov–Jun 2019–20; Aug–Sep 2020; Dec–Mar 2020–21	14	Yes	1438
Lucknow	Sep 2019–Jul 2021	-	23	Yes	23,187
Kodaikanal	May 2019–Jul 2020	Sep–Nov 2019; Feb–Jun 2020	7	No	377
Dindigul	Aug 2019–Jul 2020	Jan 2020; Apr 2020	10	No	1202
Nainital	Jun–Jul 2018	-	2	No	285
Total	-	-	143	-	66,210

**Table 2 animals-12-02000-t002:** Number of male and female dogs sterilised and sex ratio at each programme site.

Project Location	*n*	Totals		Sex Ratio (Males per Female)
		All Male	All Female	
Dehradun	20,291	9350	10,941	0.85
Vadodara	19,496	8907	10,589	1.19
Mussoorie	1438	700	738	1.05
Lucknow	23,187	10,841	12,346	1.14
Kodaikanal	377	60	317	0.19
Dindigul	1202	84	1118	0.08
Nainital	285	68	217	0.31
Total	66,210	31,450	34,826	1.12

**Table 3 animals-12-02000-t003:** Health conditions recorded during the street dog’s sterilisation process at each programme site.

Programme Site	*n*	Health Conditions						
		Cryptorchidism	Ovarian Cyst	Mange	Mon-Orchidism	Pyometra	TVT	Unicornuate/Single Uterus
Dehradun	20,291	50	47	105	62	162	190	7
Vadodara	19,496	50	7	18	28	10	33	0
Mussoorie	1438	0	6	1	7	10	25	1
Lucknow	23,187	2	34	38	7	47	202	30
Kodaikanal	377	0	2	0	1	2	0	0
Dindigul	1202	0	2	2	3	4	19	0
Nainital	285	0	3	0	0	3	2	0

**Table 4 animals-12-02000-t004:** Surgery data analyses (Dehradun, Vadodara, and Lucknow).

Programme Sites	*n*	Surgery Duration Was Not Recorded (Excluded from Analysis)	Surgeries Female *n* (%)	Surgery Duration by Sex Mean (Median) Minutes		Post-Operation Fates *n* (%)	
				Male	Female	Died	Complications
Dehradun	20,365	39	10,958 (53.9)	9.12 (8)	15.22 (12)	19 (0.09)	None recorded
Vadodara	19,984	543	8884 (45.7)	6.60 (6)	12.07 (11)	23 (0.12)	77 (0.40)
Lucknow	21,739	108	9951 (46.0)	10.41 (10)	16.98 (15)	43 (0.20)	300 (1.39)

Note that this table includes pregnant females. These were subsequently removed from analyses unless otherwise stated.

**Table 5 animals-12-02000-t005:** Surgery duration association with postoperative complications and death.

Surgery Duration versus:	Male	Female
Postoperative death	27 of 31,605 (0.09%) W = 449,507, *p* = 0.623 Not significant (Bonferroni correction)	58 of 28,709 (0.20%) W = 985,691, *p* = 0.014 Significant (Bonferroni correction)
Postoperative complications	222 of 31,605 (0.70%) W = 2196,286, *p* < 0.001 Significant (Bonferroni correction)	152 of 28,729 (0.53%) W = 1,385,782, *p* < 0.001 Significant (Bonferroni correction)

**Table 6 animals-12-02000-t006:** Sex association with postoperative complications and deaths.

	*n*	of Which Female (%)	Male (%)	Female (%)	Association	
					χ^2^	*p*-Value
Postoperative death	60,314	28,709 (47.6)	27 (0.09)	58 (0.20)	13.716 (df = 1)	<0.001
Postoperative complications			222 (0.70)	152 (0.53)	7.0257 (df = 1)	0.008

**Table 7 animals-12-02000-t007:** Age- and sex-associated postoperative complications and deaths.

		*n*	of Which Pup (%)	of Which Young (%)	Pup (%)	Young (%)	Adult (%)	Association	
								χ^2^	*p*-Value
Male	Postoperative death	31,605	603 (1.9)	5811 (18.4)	1 (0.17)	1 (0.02)	25 (0.10)	4.1882 (df = 2)	0.123
	Postoperative complications				2 (0.33)	36 (0.62)	184 (0.73)	2.0441 (df = 2)	0.360
Female	Postoperative death	28,709	431 (1.5)	8996 (31.3)	0	14 (0.16)	44 (0.23)	2.488 (df = 2)	0.288
	Postoperative complication				8 (1.86)	35 (0.39)	109 (0.57)	18.242 (df = 2)	<0.001

**Table 8 animals-12-02000-t008:** Postoperative effects of sterilising pregnant females. Males and pups were excluded from the sample.

Pregnancy		Total	Survived	Died	No Complication	Complication	Mean Surgery Time (Min)
No	Number of dogs	28,474	28,416	58	28,327	147	14.75
	Percentage	95.5	99.8	0.2	99.5	0.5	
Yes	Number of dogs	1333	1333	0	1330	3	19.37
	Percentage	4.5	100.0	-	99.8	0.2	

**Table 9 animals-12-02000-t009:** Effect of surgeon experience on surgery duration and postoperative complications and deaths.

Surgeon Status	Number of Individuals	Number of Operations Performed		Operation Time (Min)			Total Postoperative Complications (%)	Total Postoperative Deaths (%)
		Total	Mean	Lower Quartile	Median	Upper Quartile		
Experienced Vet	14	35,060	2504.3	8	10	13	391 (1.12)	58 (0.17)
Trainee Vet	8	272	34	15	24	34	14 (5.15)	0
Total	22	35,332	1606	8	10	14	405 (1.47)	58 (0.16)

## Data Availability

The original contributions presented in the study are included in the article; further inquiries can be directed to the corresponding authors.

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
