# Peer review of "Technology for Improving Street Dog Welfare and Capturing Data in Digital Format during Street Dog Sterilisation Programmes"

_animals, 2022, doi:10.3390/ani12152000_

Round 1

Reviewer 1 Report

Excellent report.  I think it is important to emphasize the very low rates of death and complications to reassure any potential critics of such programs. The analysis of surgery outcomes and duration offers some good insights into large scale spay/neuter programs.

Author Response

Dear Reviewer,

Many thanks for reviewing the manuscript and for your feedback. I have added a paragraph under discussion (line 436 - 441) which address your concerns regarding the discussion of low post-operative complications and deaths.

Warm Regards,

Dr Amit Chaudhari

Reviewer 2 Report

The manuscript "Technology for improving street dog welfare and capturing data in digital format during street dog sterilization programs" shows how useful technology can now be in various areas of managing populations of animals of various species. The authors describe the operation of the HSIApps application available for  a mobile phone for the location of the catch site of free-living dogs, so that after sterilization and vaccinations they can be released at the catch site.

In the countries of the European Union, the USA and Canada, this is not a problem as there are no free-living dogs and if a dog is left in the street they are caught and sent to a shelter.

In countries like India, free-living dogs exist and their rights are guaranteed by the Animal Birth Control Rules of 2001 (ABC Rules 2001) (is a specific law that provides legal protection to street dogs against relocation, removal, killing and poisoning).

However, free-living dogs pose a risk to humans, as the authors describe in lines 72-94. Therefore, the only available method is birth control. The work is well written, graphs and tables show the results, and screenshots from the phones of the operation of the application give an insight into how it works.

 However, the Results section should be improved.

Line 301 – „The methods used to catch dogs across the cities analysed are shown in Figure 3.” It would sound better: The methods used to catch dogs across the cities analysed there are: Hand, Net, Owner, Scruff, Trap (Figure 3).

 Line 305 „Table 2 shows the ratio of male to female dogs caught” should be changed to: the ratio of the caught males and females varied from 0.08 to 1.19 (table 2)

It is definitely better to read when you know what's going on than referring directly to a chart or table. The same is true for the following lines (306-307) and Figures 4, 5, 6, 7.

I am also wondering about fewer caught males than females. One male can leave much more offspring than one female, the castration procedure is also much simpler for males, and safer, because there is no break in the abdominal wall and no entry into the body cavity, as it is the case in females. There are also significantly fewer postoperative complications in males, as the authors write in Results lines 359-362. I Discussion lines430-436 - this is an argument to sterilize more males. The authors rightly state that line 462: It is possible that some females spayed as part of this sterilization program were not spayed at the optimal time - because the sterilization procedure can be safely performed during anoestrus and it is difficult to estimate in free-living dogs.

I have one more comment regarding the tagging information, which I suggest to complete in the text. Were the caught dogs tagged after being captured with a chip, and visually marked, so as to avoid catching the same animals again? While in males the lack of testicles is visible even without the need to catch the individual, there are no indications in females whether a female dog was sterilized and it turns out only on the operating table, because the ultrasound does not always give an unambiguous answer. In some countries, when sterilizing free-living domestic cats, one ear (e.g. left) is incised and thus it can be known that the cat is sterilized.

 The manuscript could be accepted for publication in the journal Animals after minor revision.

Author Response

Dear Reviwer,

Many thanks for your suggestions and feedback! Attached point by point response.
